# Effect of Stool Sampling on a Routine Clinical Method for the Quantification of Six Short Chain Fatty Acids in Stool Using Gas Chromatography–Mass Spectrometry

**DOI:** 10.3390/microorganisms12040828

**Published:** 2024-04-19

**Authors:** Tarek Mahdi, Aurore Desmons, Pranvera Krasniqi, Jean-Marc Lacorte, Nathalie Kapel, Antonin Lamazière, Salma Fourati, Thibaut Eguether

**Affiliations:** 1Hôpital Pitié Salpêtrière-Charles Foix, AP-HP, Service de Biochimie Endocrinienne et Oncologique, 75000 Paris, France; 2Sorbonne Université, Inserm, UMR_S 1166, Research Institute of Cardiovascular Disease, Metabolism and Nutrition, 75000 Paris, France; 3Centre de Recherche Saint-Antoine, Sorbonne Université, INSERM, AP-HP, Département Metomics, Hôpital Saint Antoine, 75000 Paris, France; 4Hôpital Pitié Salpêtrière-Charles Foix, AP-HP, Service de Coprologie Fonctionnelle, 75000 Paris, France; 5Université Paris Cité, Inserm, UMR_S 1139, 75000 Paris, France; 6Paris Center for Microbiome Medicine, Federation Hospitalo-Universitaire, 75000 Paris, France

**Keywords:** gut microbiome, metabolomic, short chain fatty acids, GC-MS, stool sampling

## Abstract

Short chain fatty acids (SCFAs) are primarily produced in the caecum and proximal colon via the bacterial fermentation of undigested carbohydrates that have avoided digestion in the small intestine. Increasing evidence supports the critical role that SCFAs play in health and homeostasis. Microbial SCFAs, namely butyric acid, serve as a principal energy source for colonocytes, and their production is essential for gut integrity. A direct link between SCFAs and some human pathological conditions, such as inflammatory bowel disease, irritable bowel syndrome, diarrhea, and cancer, has been proposed. The direct measurement of SCFAs in feces provides a non-invasive approach to demonstrating connections between SCFAs, microbiota, and metabolic diseases to estimate their potential applicability as meaningful biomarkers of intestinal health. This study aimed to adapt a robust analytical method (liquid–liquid extraction, followed by isobutyl chloroformate derivatization and GC–MS analysis), with comparable performances to methods from the literature, and to use this tool to tackle the question of pre-analytical conditions, namely stool processing. We focused on the methodology of managing stool samples before the analysis (fresh stool or dilution in either ethanol/methanol, lyophilized stool, or RNAlater^®^), as this is a significant issue to consider for standardizing results between clinical laboratories. The objective was to standardize methods for future applications as diagnostic tools. In this paper, we propose a validated GC–MS method for SCFA quantification in stool samples, including pre- and post-analytical comparison studies that could be easily used for clinical laboratory purposes. Our results show that using lyophilization as a stool-processing method would be the best method to achieve this goal.

## 1. Introduction

The gut microbiota is a complex ecosystem of more than 1000 microbial species and a population of up to 10^14^ microorganisms (mainly bacteria, but also fungi, viruses, and protists) [1], which produces a wide range of metabolites that interact with the host’s cells and, thus, influence physiological processes in the colon. In recent years, numerous studies have confirmed that the gut microbiota provides the host with several functions relevant to human physiology, particularly metabolic and immunological homeostasis [2,3,4]. Therefore, the human intestine and symbiotic microbes may be considered as super organisms.

Almost all of these contributions are made via the production of gut microbiota-derived bioactive metabolites, and microbial metabolites produced through the fermentation of undigested food. These metabolites are easily accessible for the host’s cells, acting locally in the intestine, where they may also be absorbed and influence an individual’s overall biochemistry, thereby eliciting systemic effects [5]. Thus, they contribute to the metabolic phenotype of the host and may influence the disease risk.

Short chain fatty acids (SCFAs) are the largest group of metabolic nutrients. They are mainly produced in the caecum and proximal colon through the bacterial fermentation of undigested carbohydrates that have escaped digestion in the small intestine. They may also be generated from protein and amino acid decomposition [6]. SCFAs are carboxylic organic acids with an aliphatic tail, consisting of one to six carbons, of which acetate (C2), propionate (C3), and butyrate (C4) are the most abundant (≥95%) [7,8,9]. The type and quantity of produced SCFAs depend on the amount of available substrates for microbiota digestion and the composition of the microbiota itself. However, the respective molecular ratio of SFCAs, 60:20:20 (acetate/propionate/butyrate), is relatively constant in the colon and in human feces [8].

Assuming that 50–60 g of carbohydrates reach the colon per day, the production of SCFAs has been estimated to be 400–600 mmol/day [8]. The majority of SCFAs (up to 95%) undergo rapid and effective absorption via the colon, contributing to 10% of the human daily energy intake [10], the remaining (approximately 5%) being available in feces. There is increasing evidence for the critical role of SCFAs in health and homeostasis. Microbial SCFAs, namely butyric acid, serve as a principal energy source for colonocytes, and their production is essential for gut integrity because of the associated regulation of the luminal pH, mucus production, and the effects on mucosal immune function [11]. Acetic and propionic acid, the main SCFAs absorbed, may directly modulate the host metabolic health through a range of tissue-specific mechanisms related to appetite regulation, thus leading to systemic effects, energy expenditure, glucose homeostasis, and immunomodulation [12]. Finally, a direct link has been proposed between SCFAs (qualitatively and quantitatively) and some human pathological conditions, such as inflammatory bowel disease, irritable bowel syndrome, diarrhea, and cancer [13,14,15]. The variation of SCFAs levels in patients compared to healthy controls raised interest in their use as potential biomarkers in a wide array of clinical settings. For instance, in patients with multiple sclerosis the levels of propionate were lower in patients’ serum compared to healthy controls [16,17]. In this study, a positive correlation of circulating T follicular regulatory cells and T follicular helper cells with propionate serum levels was demonstrated. Decreased levels of SCFAs in patients stools were also investigated in inflammatory bowel disease [18] and it was shown that supplementation by butyrate was an effective tool in attenuating colitis [19,20]. This shows that quantification of SCFAs is developing as a central element in the diagnosis of some diseases as well as a potential companion marker for the use of SCFAs in therapeutics.

This highlights the need for the good qualitative and quantitative detection of SCFAs adapted for use in clinical laboratories daily, which can thus provide a comprehensive understanding of the functional roles of SCFAs in the human body. Due to the inaccessibility of the human proximal colon for direct investigation and the rapid absorption of SCFAs in the colonic lumen, it is challenging to quantify SCFA production rates. Fortunately, measuring SCFAs in feces can provide a non-invasive approach to demonstrating connections between SCFAs, microbiota, and metabolic diseases to estimate their potential applicability as meaningful biomarkers of intestinal health.

Numerous groups have published methods to quantify SCFAs in fecal samples using liquid chromatography–mass spectrometry (LC–MS/MS) [21,22,23], gas chromatography with flame ionization detection (GC–FID) [24,25], or gas chromatography–mass spectrometry (GC–MS) [26,27,28,29]. GC–MS, recognized for its high sensitivity and robust separation capabilities, is a well-established technique, but is accompanied by the need for derivatization. GC–FID, also dependent on derivatization, while offering simplicity and cost-effectiveness, may fall short in terms of the sensitivity for detecting SCFAs at lower concentrations, thus limiting its applicability in specific research contexts. Conversely, LC–MS, known for its versatility, high sensitivity, and reduced derivatization requirements, is a promising alternative, especially when dealing with non-volatile SCFAs in complex sample matrices; however, this is limited by the somewhat harsh pre-analytical conditions necessary for robust SCFA dosage (i.e., mobile phase containing 1.5 mM hydrochloric acid), as well as poor chromatographic separation and feeble ionization in electrospray (ESI) due to their hydrophilicity [21,30]. The decision regarding the most suitable method hinges on the specific objectives of the research, including the targeted SCFAs, the required sensitivity, and the available resources. Researchers must carefully consider these factors to align their methodological choices with the goals of their studies, ensuring the precision and reliability of their findings.

Here, we chose to use GC–MS because of the limitations stated above with LC–MS that we deemed unsuitable for our project [21,30]. Indeed, this study aimed to adapt a robust analytical method [21] that can be used to tackle the question of pre-analytical conditions in order to make it applicable in a clinical setting for which simple procedures are needed. To our knowledge, this is the first time such a large comparison between the different options in sample management has been made. Hence, we focused on stool processing and, more specifically, the methodology for managing stool samples before analysis (fresh stool or dilution in either ethanol/methanol, lyophilized stool, or RNAlater^®^). Indeed, these pre-analytical conditions are significant issues which are sometimes overlooked in favor of the analytical performance. However, they are just as important as analytical data in considering the standardization of results between clinical laboratories, allowing easy comparisons between basic research results, and providing clinicians with the proper procedure to follow in order collect stool samples. Here, we propose a validated GC–MS method for SCFA quantification in stool samples, slightly modified from the work of Furuhashi [21]. It includes the three main SCFAs (acetate, propionate, and butyrate) and three minor ones (isobutyrate, valerate, and isovalerate). This allowed us to compare pre- and post-analytical conditions that should be used to improve the current clinical procedures in stool management.

## 2. Materials and Method

### 2.1. Chemicals

Isobutanol HPLC grade (99% purity) was purchased from Alfa Aesar (Haverhill, MA, USA). The three internal standards (IS) were propionic acid d5 (96% purity), butyric acid d8 (96% purity), and 3-methyl pentanoic acid (>98% purity), and were all purchased from Sigma Aldrich (St. Louis, MO, USA). Additionally, pyridine anhydrous (99.8% purity), isobutyl chloroformate, chloroform HPLC grade (99.8% purity), cyclohexane HPLC grade, and the boiling chip granules of 2–8 mm were purchased from VWR chemicals (Radnor, PA, USA); 99.99% pure sodium hydroxide pellets were purchased from Merck (Burlington, MA, USA). The stock solution of SCFAs at 10 mM was purchased from Sigma Aldrich (St Louis, MO, USA). Mili Q water was produced using a Millipore Milli-Q system with LC-PAK Polisher/Millipak Express 40 and was purchased from Merck Millipore (Burlington, MA, USA).

### 2.2. Preparation of Stool Samples

Fecal samples used for the development of the SCFA method were either obtained directly from healthy volunteers, or were received from the routine workload of the laboratory for the follow-up of patients. The study using stool residues was approved by the French Public Health Organization (CSP-article L1121-3, amended by the law n°2011–2012, 29 December 2011-article 5), and written informed consent was provided by all volunteers. Fecal samples were collected from 28 healthy volunteers with the following criteria: (i) age ranging from 20 to 65 years; (ii) no digestive pathologies; (iii) BMI < 30; and (iv) no antibiotics in the previous three months. When the information was available, the probiotic and butyric acid intake were considered as exclusion criteria. Samples were then sent to the laboratory within 2 h of collection. For the determination of SCFAs in patients with inflammatory bowel diseases, aliquots were taken from 27 anonymized samples (ulcerative colitis (*n* = 4) and Crohn’s disease (*n* = 23), and sent to the lab for calprotectin measurements. Patients were then stratified according to the calprotectin measurements (i.e., < or > to 50 µg/g recognized as the cut-off for mucosal inflammation). For the preparation of lyophilized fecal samples, fresh stools were frozen at −80 °C for a minimum of 24 h, then subjected to freeze-drying (Lyovapor Buchi L-200, Cosmos, France). Vials were stored at +4 °C or at −80 °C prior to analysis. Fresh fecal samples were kept at 4 °C or −80 °C right after collection, or after the addition of RNAlater, ethanol, or methanol.

### 2.3. SCFA Extraction and Derivatization Method

Five to ten mg of lyophilized stools were weighed in Eppendorf tubes. Normalization was completed using the stool weight, and the final result was expressed in µmol/g of lyophilized feces. Furthermore, 10 µL of the internal standard solution (357 mmol/L in isobutanol) (pre extraction spiking) was added, followed by 1 mL of isobutanol at 10% (in water). The mixture was vortexed for 1 min and centrifuged at 14,000× *g* for 5 min at room temperature. Moreover, 675 µL of the supernatant was transferred into a new tube, either with or without the addition of an internal standard solution (post extraction spiking), and then 125 µL of NaOH at 20 mmol/L and 400 µL of chloroform were added. The resulting solution was vortexed for approximately 10 sec, and then centrifuged at 14,000× *g* for 2 min at room temperature. Additionally, 400 µL of the upper phase was moved to a new tube, and 80 µL of isobutanol, 100 µL of pyridine, and 70 µL of miliQ water were added. To minimize foam formation, boiling chips were added prior to vortexing. Sample derivatization was achieved by adding 50 µL of isobutyl chloroformiate. In order to free the produced gas, the lid was left open for 1 min. Furthermore, 150 µL of cyclohexane was then added, and the final sample was centrifuged at 14,000× *g* for 2 min. The supernatant was then transferred into GC glass vials.

### 2.4. GC-MS Conditions

Our method was validated using an Agilent 8890 GC system in tandem with an Agilent 5977B MS (Agilent technologies, Santa Clara, CA, USA). The gas chromatography (GC) conditions were as follows: GC column, HP-5MS (Agilent), 30 m × 250 µm × 0.25 µm; column temperature: 50 °C (5 min)–150 °C (0 min) at 5 °C/min, 150 °C (0 min)–325 °C (1 min) at 40 °C/min; injection port temperature = 260 °C; flow rate = 1 mL/min, injection volume = 1 µL; Split 1:50, carrier gas: helium (99.999% purity). The ionization source and quadrupole temperatures were set to 230 °C and 150 °C, respectively. Quantifications from chromatograms were calculated using Quant-my-way software part of MassHunter quantitative analysis Version 10.2 Build 10.2.733.8 (Agilent Technologies, Santa Clara, CA, USA). The SCFAs quantified and the internal standards used are listed with their retention times and their monitor ions in Table 1.

### 2.5. Method Validation

Our laboratory is accredited by COFRAC, following the norm ISO 15189 [31]. Hence, the method validation was based on the recommendations of NF EN ISO 15189 criteria and the international guidelines dedicated to LC-MS/MS methods [32,33,34,35].

#### 2.5.1. Extraction Recovery (ER%) and Matrix Effect (ME%)

The extraction recovery (ER%) and matrix effect (ME%) were evaluated as follows: Equation (1) describes the extraction recovery assessment, while Equation (2) illustrates the matrix effect evaluation.

The extraction recovery was estimated by comparing spiked QC feces samples before and after the extraction at two different concentrations.
(1)$$ER\%=\frac{analyte\;peak\;area\;spiked\;before\;extraction}{analyte\;peak\;area\;spiked\;after\;extraction}\times 100$$

The matrix effect evaluation was achieved in two different concentrations by comparing spiked QC feces samples following extraction to the neat standard solvent solutions at the same concentration.
(2)$$ME\%=\frac{Mean\;peak\;area\;of\;the\;spiked\;analyte\;in\;matrix-Mean\;peak\;area\;of\;the\;endogenous\;analyte\;in\;matrix}{Mean\;peak\;area\;of\;the\;analyte\;in\;water}\times 100$$

#### 2.5.2. Linearity and Sensitivity (LOD, LOQ)

The quantification of the endogenous SCFAs was achieved via a calibration curve. The calibration curve was built with eight different levels of calibrants, and was analyzed based on the response factor (analyte area/IS area).

Sensitivity was estimated as a limit of detection (LOD) and a limit of quantification (LOQ). LOD and LOQ were calculated experimentally, based on signal/noise ratios of 3:1 and 10:1, respectively, by analyzing spiked and extracted feces samples (Table 1).

#### 2.5.3. Precision and Accuracy

Accuracy and precision were estimated from three feces pools used as QC samples with three different concentrations. Intra-day precision and accuracy were estimated by analyzing a series of the three QC feces samples in *n* = 20 replicates within the same day. Inter-day precision and accuracy were evaluated via analyzing a series of the same three QC feces samples in *n* = 10 replicates across a period of 10 days.

#### 2.5.4. Stability

To evaluate the impact of stool collection modalities on acetate, propionate, isobutyrate, butyrate, isovalerate, and valerate levels, a single stool was collected using four different storage conditions (fresh stool or dilution in either ethanol, methanol, or RNAlater^®^, Thermo Fisher Scientific, Waltham, MA, USA). After homogenization, the same amount of stool was weighed and a factor corresponding to the initial dilution was applied to the final result to allow direct comparison between diluted samples and fresh stool samples. Extraction was then performed as previously described.

To evaluate the stability of the six SCFAs at room temperature, quantification was performed on a single stool sample. Aliquots from the same stool were homogenized and stored at −80 °C from T0 until T8 h every 2 h.

To assess the impact of freeze–thawing on the measurement, six SCFAs were quantified after two and five freeze–thaw cycles, and then compared to the fresh and lyophilized stool samples stored at −80 °C.

To evaluate the stability of SCFAs at +4 °C, three independent lyophilized stools were stored at +4 °C, and the SCFAs were quantified at days 12, 33, 60, and 130 and then compared to the reference value obtained at day 1.

As glycerol 10% solution is frequently used as a cryoprotectant in stool collections, we evaluated the impact of a 1:4 dilution (*w*/*v*) of the samples. The comparison was performed on fresh, lyophilized, and 10% glycerol-diluted stools using three independent human stools.

Finally, to evaluate the stability of the SCFAs following the extraction and derivatization of either fresh or lyophilized samples, we compared extracts immediately after production and again after seven days of storage at +4 °C. The stability of stool samples at room temperature was also assessed following extraction and derivatization by measuring the concentration of the same six SCFAs. We used ten different lyophilized stools; measurements were made before and after a 24 h stay on the autosampler.

### 2.6. Statistical Analysis

Results are expressed as mean ± SD. Statistical analyses were performed with a Mann–Whitney test through the use of Graph Pad Prism software v.6.0 (La Jolla, CA, USA). The statistical difference was considered significant when *p* < 0.05.

## 3. Results and Discussion

### 3.1. Method Validation

We followed the required specifications of the October 2019 FDA (Food and Drug Administration) guidelines. The linearity ranges for each of the six SCFAs were established using calibration curves prepared from a stock solution. All calibration curves reached an acceptable coefficient of determination (R^2^ > 0.990) (Table 1). The concentration ranges and LOQs of each compound ranged from 0.76 to 1500 nmol and 0.76 to 18.64 nmol, respectively, as detailed in Table 1. Intraday and interday precision showed satisfying results for all quantified SCFAs, with a standard deviation ranging from 3% to 10% for interday precision, and 1 to 5% for intraday precision (Table 2). The extraction recovery and the matrix effect are expressed in Table 3. The matrix effect was acceptable (between 80 and 120%) for all SCFAs, ranging from 93 to 112%, except for valerate and isovalerate, for which we observed percentages slightly above the acceptable limits (Table 3). The extraction recoveries were also acceptable (between 80 and 120%) for acetate, propionate, and butyrate, with the results falling between 81 and 115%. However, a moderate increase was observed for isobutyrate, valerate, and isovalerate, with the highest recovery rate of 129%, thus suggesting a risk of overestimation. These results could be explained via the difference in abundance (lower concentrations) between acetate, propionate, butyrate, and the other measured SCFAs.

### 3.2. Comparison of Sample Storage

Significant variations in SCFA concentrations were found depending on the four storage conditions (Figure 1 and Table 4). Storing stools in ethanol or RNAlater^®^ led to a mean concentration of SCFAs that was lower than 20% of the concentration found in fresh stools. Storage in methanol led to higher concentrations, with values varying from 44 to 54% for acetate, propionate, butyrate, and valerate, and from 62 to 68% for isovalerate and isobutyrate when compared to the concentrations measured in fresh stools. Stool collection conditions are therefore a significant issue for SCFA quantification regarding the assessment of reliable results. Ethanol, methanol, and RNAlater^®^ storage are currently used for stool metabolite studies. Although our results need to be confirmed using more samples, the decrease in SCFA concentrations quantified in stools stored in those environments when compared to fresh stools demonstrated that these modalities should be rejected for SCFA measurements in feces. In contrast, we confirmed that fresh stool samples being stored at −80 °C was the best adapted storage condition. Similar results have been previously documented [29,36].

### 3.3. Impact of Lyophilization and Glycerol Addition

We found equivalent or higher concentrations of all measured SCFAs in lyophilized stools than in fresh stools. The variations in concentration ranged from −7 to 165% for stools in glycerol. SCFA concentrations variations ranged from −9 to 42% of the concentrations found in fresh stools, and similar findings were previously described by Reygner et al. [36]. Stool B was an exception, where we found a decrease in butyrate, isobutyrate, isovalerate, and valerate, representing −43 to −25% of the concentration in fresh stools. These results are summarized in Figure 2 and Table 5. Hence, SCFA concentrations appeared to be increased in samples preserved via freeze-drying. Controversial results using this process have already been described, as Zheng and colleagues [37] found lower concentrations, but Reygner et al. and Ueyama et al. found similar results [29,36]. Our results, together with the fact that the freeze-drying procedure allowed for the elimination of bias resulting from the hydration levels in the stool, make lyophilized stools an interesting tool for SCFA quantification.

### 3.4. Impact of Freeze–Thawing and +4 °C Stool Storage

The measurements of SCFAs in lyophilized stools after two freeze–thaw cycles showed variations ranging from −8 to +1% for acetate, propionate, butyrate, and isovalerate concentrations, and a loss of 32 and 35% for isobutyrate and valerate, respectively. After five cycles of freezing and thawing, acetate, propionate, and butyrate concentrations in lyophilized stools decreased from 12 to 29%, whereas the loss of isovalerate, isobutyrate, and valerate concentrations varied from 48 to 50%. Notably, the decrease in SCFA concentrations was more pronounced in fresh stools than in lyophilized stools, as shown in Table 6. Concerning stability levels following +4 °C storage, variations were lower than 15% for acetate, propionate, and butyrate at day 60 in all samples. Similar variations were observed for isobutyrate, isovalerate, and valerate for two of the three tested samples. The remaining samples showed decreases in concentration, ranging from 35 to 51% per SCFA. SCFA concentrations at day 130 displayed variations from 17 to 78%, except for butyrate, which was remarkably stable, with variations between 2 and 12%. The percentages of variations for each SCFA are related to the concentrations of the same SCFA on day 1 (Table 7 and Figure 3). Our results follow the results of Ueyama et al. [29], who described SCFA concentrations as stable for at least three days in lyophilized stools stored at room temperature (20–25 °C). Variations during freeze–thaw cycles showed that storing lyophilized stools at −80 °C is not a good option for optimal stability, involving a risk of moistening (Table 6). According to our results and previous results [29], storing lyophilized stools at +4 °C for up to two months may be considered an optimal condition for SCFA quantification.

### 3.5. Stability of SCFA after Stool Collection

The quantification of SCFAs following stool collection showed a slight variation in concentrations of all SCFAs after 2 h of storage at room temperature, with a decrease ranging from −29 to −11%, while a slight increase in the SCFA concentration was observed over time. After 6 and 8 h of storage at room temperature, SCFA concentrations ranged from −14 to 2% and from −5 to 21% of the reference value, respectively (Table 8 and Figure 4). All the different percentages are expressed as a ratio to the concentrations of SCFAs quantified in the stools that were stored at −80 °C immediately following issuance.

Our results showed optimal stability for all SCFAs quantified after 6 h of storage at room temperature.

The SCFA stability assays at room temperature over time presented the same pattern for all SCFAs, independent of their concentrations. Stability was acceptable even after 8 h of storage at room temperature. As far as we know, no previous studies have examined the stability of SCFAs following issuance. Fecal SCFA quantification may thus be quantified in a sample even after 6 h at room temperature, allowing easy transport in routine practice.

### 3.6. Post-Extraction and Derivatization Stability

The stability of SCFAs in stool extracts stored at +4° was satisfactory, ranging from −13 to 3% for lyophilized stool following 7 days of storage. Similar variations were noticed for fresh stool from −14 to 3%, as described in Table 9 and Figure 5. The room temperature storage of stool extracts also showed satisfactory stability following 24 h of storage, ranging from −11% to 15% (Figure 6). The stability of the samples at +4 °C and room temperature after extraction and derivatization may be a considerable asset for routine laboratory work, as it allows multiple measurements of samples and controls to be taken without affecting the accuracy of the results.

### 3.7. Application with Clinical Samples

Our developed method was then tested on lyophilized stools of patients diagnosed with intestinal bowel disease (IBD) (*n* = 27), and compared to those from non-affected individuals (*n* = 28). All samples were also tested for calprotectin in order to stratify IBD patients according to published cutoff values (<50 and >50 µg/g of stool). No significant differences were observed in IBD patients when compared to non-affected individuals. Stratification based on fecal calprotectin levels did not improve significance (Figure 7 and Table 10). The heterogeneity of our IBD cohort, made up of UC and CD patients undergoing multiple therapeutic approaches and with different severity stages, makes the interpretation of results more challenging. Despite this interindividual variability, the median butyrate level decreased in patients with increased calprotectin level (nonsignificant), as also observed by Huda–Faujan et al. [38]. In their study, propionate and butyrate concentrations were significantly lower in an IBD group (CD patients *n* = 2 and UC patients *n* = 6) than in healthy subjects (*n* = 50). The differences in results with our study could be explained by the proportion of UC/CD patients in the IBD group. In another study, Lemay et al. [39] evaluated the SCFA concentrations in the lyophilized stools of a child cohort with CD (*n* = 12) and UC (*n* = 10), treated with exclusive enteral nutrition (EEN) or corticosteroid (CS) therapy. Butyrate was significantly increased (2.6-fold higher) in CD patients when compared to UC patients. UC is often characterized by the extensive luminal inflammation of the colon, which leads to lower colonic luminal bacterial populations. This differs from CD, which may not have extensive colonic inflammation at the same level, but generally includes the inflammation of the upper gastrointestinal tract and the small intestine. *Roseburia intestinalis* and *Faecalibacterium prausnitzii,* essential butyrate-producing intestinal bacteria, have been described to be present in lower proportions in UC patients, and their abundances are inversely correlated to disease severity [18]. In addition, UC patients are well known to have faster transmission times, thus reducing microbial fermentation in the colonic intestinal epithelium [40]. Studies with large, well-characterized cohorts of UC and CD patients compared to healthy subjects should be considered to improve SCFA interpretation, as shown by two studies by Kaczmarcsyk et al. [41,42]. Indeed, the group compared SCFAs in stools kept fresh at −80 °C, and then used evaporation to dry and grind them before extraction in 35 patients [41]. They did not show any difference between CD and UC patients, irrespective of disease activity or the control group, although the propionic acid levels correlated with trimebutine intake. One year later, using the same technique in another study with 61 patients and 16 controls [42], a study showed that there were significant decreases in acetic, butyric, isovaleric, and valeric acids, as well as an increase in lactic acid, thus confirming that the number of patients involved in the study is critical in demonstrating a significant difference between groups. Finally, SCFAs are low in IBD patients, especially during flare-ups, due to a lower number of bacteria-producing SCFAs in their microbiome, with involvement in physiopathology that is not yet understood [14]. This requires more clinical studies to better understand the pathophysiological mechanisms, and how SCFAs could help with the therapeutic management of IBD patients. This will only be possible if preanalytical and analytical methods are standardized.

## 4. Conclusions

In our study, we developed a validated method for the simultaneous determination of six SCFAs (acetate, propionate, isobutyrate, butyrate, isovalerate, and valerate) via the use of GC–MS. Although it is possible to process samples collected in RNAlater or ethanol, the best results were obtained when fresh stool samples were either stored at −80 °C or lyophilized and stored at 4 °C. Intraday and interday precision results obtained from lyophilized stools were satisfactory for all quantified SCFAs. The matrix effect, extraction recovery, and extraction process efficiency were also satisfactory for acetate, propionate, and butyrate, but less so for isobutyrate, isovalerate, and valerate, which is likely due to their lower concentrations within the stool samples. Using lyophilized stools as a main quantification matrix allowed for the elimination of bias relating to the stool hydration degree. Moreover, lyophilized stools offered better homogenization and conservation stability for SCFA quantification. The major strengths of our study are the pre- and post-analytical analyses, which involved evaluating different storage temperatures and multiple storage solvents. Our results regarding SCFA stability following issuance suggest SCFA stability even after 6 h of transport at room temperature. These results could considerably simplify sample delivery to labs. Although SCFA concentrations showed no significant differences between IBD patients and healthy subjects, we were able to develop and validate a sensitive, specific, and reproducible quantification technique. This method could be a valuable asset for future studies of SCFAs. Thus, we recommend a GC−MS method for the SCFA quantification of lyophilized stool samples for clinical purposes.

## Figures and Tables

**Figure 1 microorganisms-12-00828-f001:**
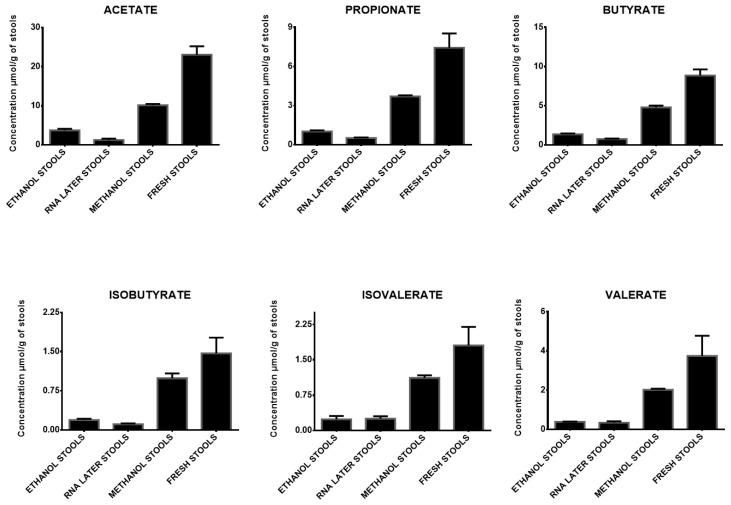
Impact of the storage medium on SCFA concentration in human stools. Concentrations of 6 short chain fatty acids (acetate, propionate, isobutyrate, butyrate, isovalerate, valerate), extracted from a unique human stool sample collected in four different storage conditions (ethanol, RNAlater, methanol, fresh) and stored at −80 °C. Quantifications were made in triplicate (*n* = 3) for each condition.

**Figure 2 microorganisms-12-00828-f002:**
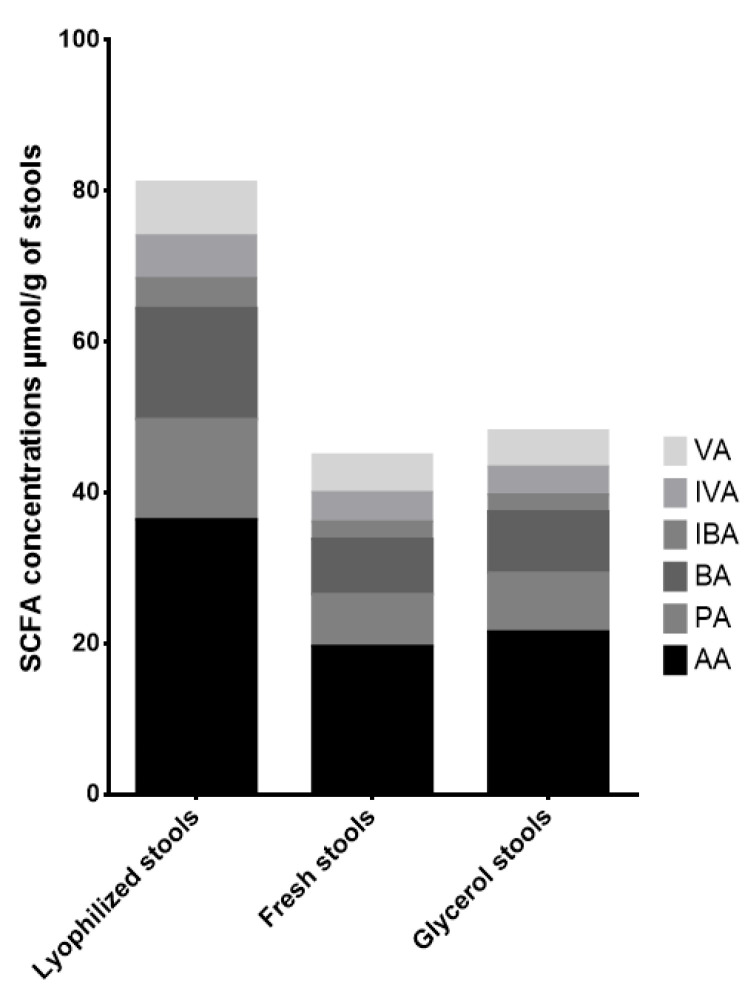
SCFA concentrations in stools in function of storage processes. Concentrations of 6 short chain fatty acids (acetate, propionate, isobutyrate, butyrate, isovalerate, valerate) quantified on the fresh, lyophilized, and 10% glycerol-added stools of three independent human stools stored at −80 °C. The measurements were made in triplicate (*n* = 3) for each condition. AA: acetate; PA: propionate; BA: butyrate; IBA: isobutyrate; IVA: isovalerate; VA: valerate.

**Figure 3 microorganisms-12-00828-f003:**
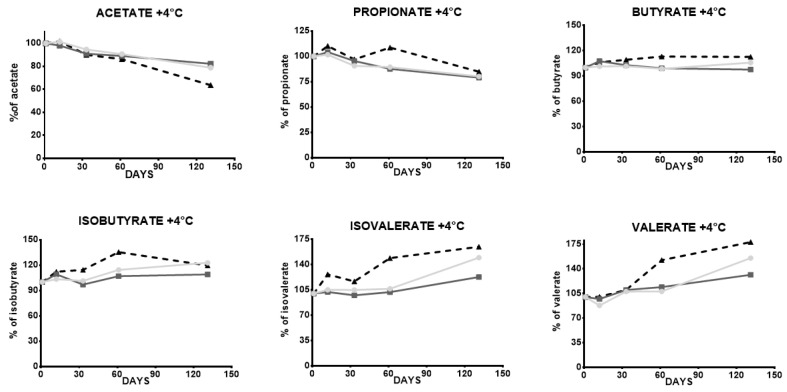
SCFAs stability assay at +4 °C. The mean percentages per day (12, 33, 60, 130) of 6 short chain fatty acids (acetate, propionate, isobutyrate, butyrate, isovalerate, valerate), extracted from three independent lyophilized stools, stored at +4 °C, measured in triplicate (*n* = 3). Stool 1: light gray dots; Stool 2: dark gray scares; Stool 3: dark triangles. The variation percentages of each SCFA are related to the concentrations of the same SCFA at day 1. Day 1 corresponds to the first quantification of SCFA made on the sample.

**Figure 4 microorganisms-12-00828-f004:**
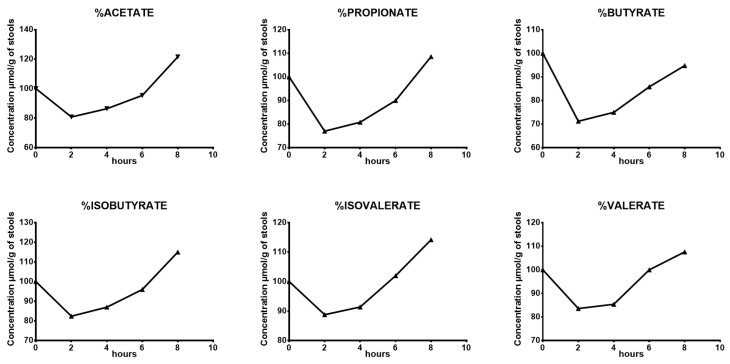
SCFA stability assay at room temperature over time. Mean percentages of 6 short chain fatty acids (acetate, propionate, isobutyrate, butyrate, isovalerate, valerate) left at room temperature over time (2 h, 4 h, 6 h, 8 h). All the variation percentages are expressed as a ratio to concentrations of SCFAs quantified in the stool stored at −80 °C immediately after issuance.

**Figure 5 microorganisms-12-00828-f005:**
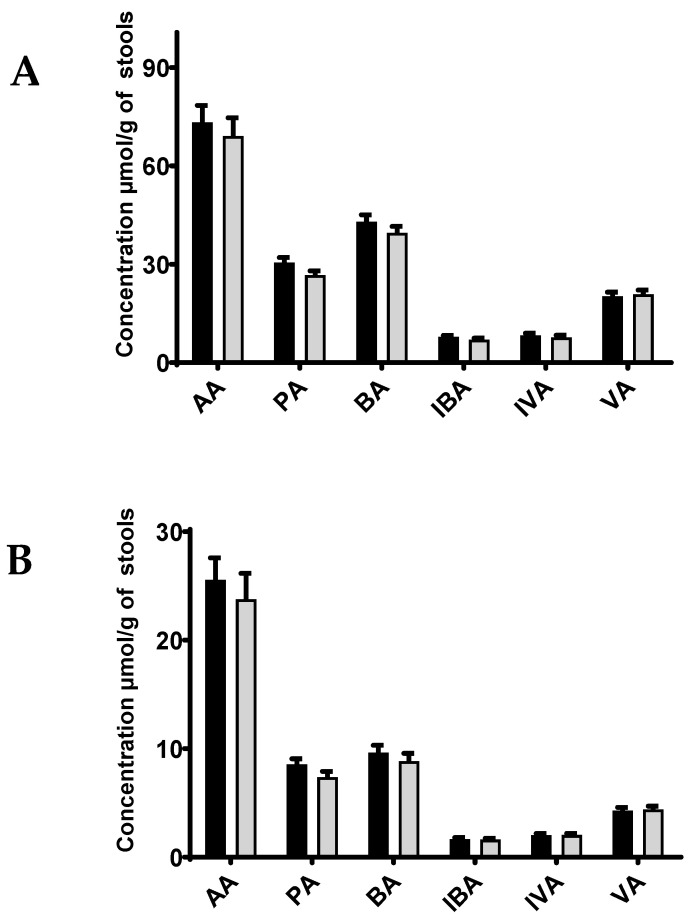
Effect of extraction and derivatization on SCFA concentrations before and after an extract storage at +4 °C for 7 days. Concentrations of 6 short chain fatty acids (acetate, propionate, isobutyrate, butyrate, isovalerate, valerate) extracted from a unique human stool ((**A**): lyophilized stool, (**B**): fresh frozen stool) before (dark bars) and after (light gray bars) 7 days of storage of the extract at +4 °C, measured in triplicate (*n* = 3). AA: acetate; PA: propionate; BA: butyrate; IBA: isobutyrate; IVA: isovalerate; VA: valerate.

**Figure 6 microorganisms-12-00828-f006:**
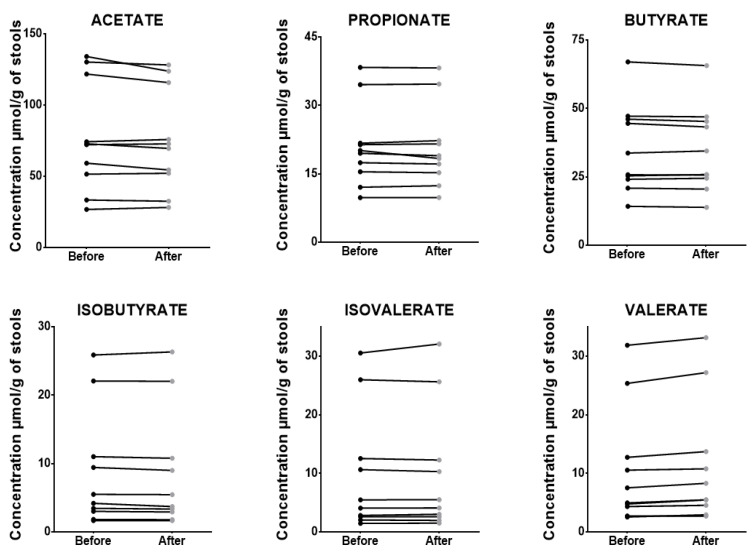
SCFAs concentrations before and after an extract storage at room temperature for 24 h. Concentrations of 6 short chain fatty acids (acetate, propionate, isobutyrate, butyrate, isovalerate, valerate) extracted from ten different human lyophilized stools (S1–S10) both before (black dots) and after (light gray dots) a 24 h stay of the extract on the autosampler at room temperature.

**Figure 7 microorganisms-12-00828-f007:**
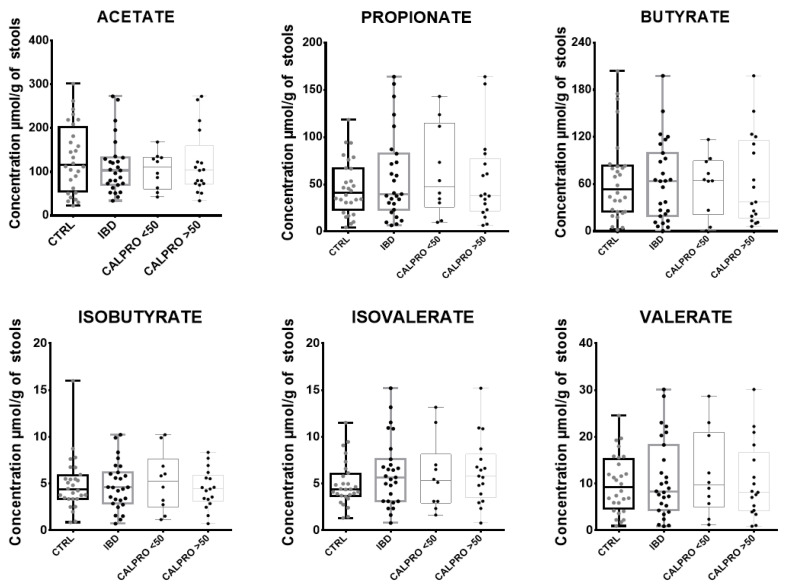
SCFAs concentrations in control group vs. IBD group. Concentrations of 6 short chain fatty acids (acetate, propionate, isobutyrate, butyrate, isovalerate, valerate), extracted from the stools of control subjects *n* = 28 (CTRL, dark box plot bars) and from the stools of inflammatory bowel disease patients *n* = 27 (IBD, dark gray box plot bars). IBD patient group was subdivided according to fecal calprotectin concentration into 2 subgroups. IBD patients with calprotectin lower than 50 µg/g of stools (CALPRO < 50, gray box plot bars) and IBD patients with calprotectin higher than 50 µg/g of stools (CALPRO > 50, light gray box plot bars).

**Table 1 microorganisms-12-00828-t001:** Monitor ions and retention times of SCFAs and their internal standards, linearity, and parameters of the calibration curves. LOD: Lower limit of detection; LOQ: Lower limit of Quantification.

	Internal Standard	Monitor Ion (M/Z)	Retention Time (min)	Calibration Range (nmol)	LOD (nmol)	LOQ (nmol)	Slope	Intercept	R²
Acetate	Acetate D4	56.1	3.86	12.89–1500	3.87	12.89	1.424 × 10^−4^	5.580 × 10^−4^	0.991
Propionate	Propianate D5	57.1	6.64	0.76–1500	0.2	0.76	4.922 × 10^−4^	6.517 × 10^−4^	0.996
Butyrate	Butyrate D8	71.1	9.80	18.64–1500	6.08	18.64	2.656 × 10^−4^	5.391 × 10^−4^	0.999
Isobutyrate	Butyrate D8	71.1	8.30	1.86–1500	0.54	1.86	4.481 × 10^−4^	5.908 × 10^−4^	0.999
Valerate	3-Methyl pentanoic acid	85.1	13.25	4.9–1500	1.9	4.9	3.051 × 10^−4^	3.630 × 10^−4^	0.995
Isovalerate	3-Methyl pentanoic acid	85.1	11.60	3.09–1500	1.2	3.09	3.623 × 10^−4^	1.043 × 10^−4^	0.994
Acetate D4		60.1	3.80						
Propionate D5		62.1	6.55						
Butyrate D8		78.2	9.95						
3-Methyl pentanoic acid		99.1	15.15						

**Table 2 microorganisms-12-00828-t002:** Intraday and interday precision. Intraday and interday precision (*n* = 20) of six short chain fatty acids (acetate, propionate, isobutyrate, butyrate, isovalerate, valerate), evaluated with three different human lyophilized stools stored at +4 °C (stool 1.2.3).

	Stools	Concentration *Intraday* (µmol/g)	SD *Intraday* (µmol/g)	CV *Intraday* (%)	Concentration *Interday* (µmol/g)	SD *Interday* (µmol/g)	CV *Interday* (µmol/g)
Acetate	1	46.22	4.59	10%	45.32	0.98	2%
	2	52.78	3.69	7%	49.16	2.47	5%
	3	105.41	3.86	4%	107.08	2.82	3%
Propionate	1	13.47	0.85	6%	12.29	0.18	2%
	2	19.74	0.68	3%	19.42	0.35	2%
	3	24.23	0.89	4%	23.06	0.64	3%
Butyrate	1	10.93	0.68	6%	9.63	0.12	1%
	2	24.23	0.84	3%	23.97	0.14	1%
	3	45.31	1.84	4%	42.97	0.63	1%
Isobutyrate	1	2.83	0.2	7%	2.62	0.06	2%
	2	1.71	0.08	5%	1.68	0.05	3%
	3	3.3	0.19	6%	3.14	0.13	4%
Valerate	1	2.97	0.19	6%	2.99	0.05	2%
	2	2.47	0.09	4%	2.81	0.05	2%
	3	4.67	0.24	5%	4.95	0.22	4%
Isovalerate	1	3.3	0.25	8%	2.8	0.05	2%
	2	1.57	0.07	5%	1.52	0.03	2%
	3	2.74	0.19	7%	2.53	0.06	3%

**Table 3 microorganisms-12-00828-t003:** Matrix effect and extraction recovery. Evaluation was assessed with two lyophilized human stools stored at +4 °C, spiked by two levels of the short chain fatty acids stock solution (18.5 and 167 nmol).

	Stools	Spiked (nmol)	Matrix Effect	Extraction Recovery
Acetate	Stool 1	167	95%	113%
		18.5	106%	91%
	Stool 2	167	109%	81%
		18.5	113%	87%
Propionate	Stool 1	167	91%	115%
		18.5	105%	88%
	Stool 2	167	93%	100%
		18.5	107%	91%
Butyrate	Stool 1	167	101%	111%
		18.5	107%	87%
	Stool 2	167	108%	94%
		18.5	112%	88%
Isobutyrate	Stool 1	167	100%	124%
		18.5	106%	97%
	Stool 2	167	93%	121%
		18.5	108%	98%
Valerate	Stool 1	167	112%	127%
		18.5	116%	96%
	Stool 2	167	109%	119%
		18.5	123%	96%
Isovalerate	Stool 1	167	115%	126%
		18.5	121%	98%
	Stool 2	167	109%	129%
		18.5	129%	96%

**Table 4 microorganisms-12-00828-t004:** Impact of storage medium on SCFA concentration in human stools. The mean concentrations and percentages of 6 short chain fatty acids (acetate, propionate, isobutyrate, butyrate, isovalerate, valerate), extracted from a unique human stool sample collected in four different storage conditions, measured in triplicate (*n* = 3) for each condition. Percentages of SCFAs in each condition are expressed in relation to the concentration of the same SCFA in fresh stools. AA: acetate; PA: propionate; BA: butyrate; IBA: isobutyrate; IVA: isovalerate; VA: valerate.

	AA	SD(AA)	PA	SD(PA)	BA	SD(BA)	IBA	SD(IBA)	IVA	SD(IVA)	VA	SD(VA)
Mean concentrations in ethanol stools	3.71	0.38	1.01	0.05	1.36	0.10	0.19	0.02	0.24	0.07	0.38	0.01
Mean percentages in ethanol stools	16%		14%		15%		13%		13%		10%	
Mean concentrations in RNA later stools	1.26	0.31	0.52	0.03	0.76	0.05	0.11	0.02	0.25	0.05	0.34	0.07
Mean percentages in RNA later stools	5%		7%		9%		8%		14%		9%	
Mean concentrations in methanol stools	10.13	0.33	3.70	0.08	4.80	0.23	0.99	0.09	1.12	0.05	2.02	0.05
Mean percentages in methanol stools	44%		50%		54%		68%		62%		54%	
Mean concentrations in fresh stools	23.04	2.17	7.44	0.63	8.85	0.79	1.47	0.17	1.81	0.39	3.75	1.03

**Table 5 microorganisms-12-00828-t005:** Influence of stool storage processes. Mean variation percentages per stool storage condition (lyophilized, glycerol) of 6 short chain fatty acids (acetate, propionate, isobutyrate, butyrate, isovalerate, valerate), quantified on lyophilized and 10% glycerol-added stools of three independent human stools stored at −80 °C, measured in triplicate (*n* = 3). The variation percentages of each SCFA in each stool sample are related to the concentrations of the same SCFA in the corresponding fresh stool.

		Lyophilized Stools	Glycerol Stools
Acetate	STOOL A	54%	0.4%
	STOOL B	84%	−0.3%
	STOOL C	117%	25%
Propionate	STOOL A	34%	−0.3%
	STOOL B	108%	−9%
	STOOL C	134%	42%
Butyrate	STOOL A	0.9%	−0.3%
	STOOL B	94%	−33%
	STOOL C	165%	37%
Isobutyrate	STOOL A	30%	0.3%
	STOOL B	96%	−25%
	STOOL C	100%	−0.1%
Isovalerate	STOOL A	10%	0.6%
	STOOL B	84%	−35%
	STOOL C	74%	−0.1%
Valerate	STOOL A	−0.7%	0.2%
	STOOL B	70%	−43%
	STOOL C	94%	14%

**Table 6 microorganisms-12-00828-t006:** Freeze–thaw stability, evaluated on a unique human stool (Fresh/Lyophilized) stored at −80 °C. Measured in triplicate (*n* = 3). *: µmol/g of lyophilized stools.

		Concentration (µmol/g of Stools)	Variation (µmol/g of Stools)	Variation %
Acetate	fresh stool	25.55		
	fresh stool 2FT	21.81	−3.74	−15%
	fresh stool 5FT	22.78	−2.77	−11%
	lyoph stool	73.29 *		
	lyoph stool 2FT	74.04 *	0.75 *	1%
	lyoph stool 5FT	64.28 *	−9.01 *	−12%
Propionate	fresh stool	8.56		
	fresh stool 2FT	6.37	−2.19	−26%
	fresh stool 5FT	6.25	−2.31	−27%
	lyoph stool	30.55 *		
	lyoph stool 2FT	28.27 *	−2.28 *	−7%
	lyoph stool 5FT	23.58 *	−6.97 *	−23%
Butyrate	fresh stool	9.63		
	fresh stool 2FT	8.05	−1.58	−16%
	fresh stool 5FT	8	−1.63	−17%
	lyoph stool	42.92 *		
	lyoph stool 2FT	44.07 *	1.15 *	3%
	lyoph stool 5FT	30.56 *	−12.36 *	−29%
Isobutyrate	fresh stool	1.66		
	fresh stool 2FT	1.12	−0.54	−33%
	fresh stool 5FT	0.93	−0.73	−44%
	lyoph stool	7.84 *		
	lyoph stool 2FT	5.36 *	−2.48 *	−32%
	lyoph stool 5FT	3.89 *	−3.95 *	−50%
Valerate	fresh stool	4.28		
	fresh stool 2FT	2.56	−1.72	−40%
	fresh stool 5FT	2.12	−2.16	−50%
	lyoph stool	20.27 *		
	lyoph stool 2FT	13.1 *	−7.17 *	−35%
	lyoph stool 5FT	10.37 *	−9.9 *	−49%
Isovalerate	fresh stool	2.02		
	freshstool 2FT	1.35	−0.67	−33%
	fresh stool 5FT	1.01	−1.01	−50%
	lyoph stool	8.29 *		
	lyoph stool 2FT	7.63 *	−0.66 *	−8%
	lyoph stool 5FT	4.35 *	−3.94 *	−48%

**Table 7 microorganisms-12-00828-t007:** SCFAs stability assay at +4 °C. The mean variation percentages per day (12, 33, 60, 130) of 6 short chain fatty acids (acetate, propionate, isobutyrate, butyrate, isovalerate, valerate), extracted from three independent lyophilized stools stored at +4 °C, measured in triplicate (*n* = 3). The variation percentages of each SCFA are related to the concentrations of the same SCFA at day 1. Day 1 corresponds to the first quantification of SCFAs made on the sample.

		12 Days	33 Days	60 Days	130 Days
Acetate	STOOL 1	2%	−5%	−9%	−21%
	STOOL 2	−2%	−9%	−11%	−18%
	STOOL 3	2%	−10%	−14%	−36%
Propionate	STOOL 1	1%	−9%	−11%	−20%
	STOOL 2	4%	−4%	−12%	−21%
	STOOL 3	10%	−3%	9%	−15%
Butyrate	STOOL 1	1%	2%	−1%	6%
	STOOL 2	8%	3%	−1%	−2%
	STOOL 3	6%	9%	13%	12%
Isobutyrate	STOOL 1	3%	1%	15%	23%
	STOOL 2	9%	−3%	7%	9%
	STOOL 3	12%	14%	35%	20%
Isovalerate	STOOL 1	5%	5%	6%	50%
	STOOL 2	2%	−2%	2%	23%
	STOOL 3	26%	17%	49%	64%
Valerate	STOOL 1	−12%	8%	8%	56%
	STOOL 2	−3%	10%	14%	31%
	STOOL 3	1%	10%	52%	78%

**Table 8 microorganisms-12-00828-t008:** SCFA stability assay at room temperature over time. Mean variation percentages per hour (2 h, 4 h, 6 h, 8 h) of 6 short chain fatty acids (acetate, propionate, isobutyrate, butyrate, isovalerate, valerate) quantified on the lyophilized stool of a unique stool stored at −80 °C measured in triplicate (*n* = 3). The variation percentages of each SCFA are related to the concentrations of the same SCFA quantified in the corresponding stool stored at −80 °C immediately after issuance.

	2 Hours	4 Hours	6 Hours	8 Hours
Acetate	−19%	−14%	−5%	21%
Propionate	−23%	−19%	−10%	9%
Butyrate	−29%	−25%	−14%	−5%
Isobutyrate	−17%	−13%	−4%	15%
Isovalerate	−11%	−9%	2%	14%
Valerate	−16%	−15%	0%	8%

**Table 9 microorganisms-12-00828-t009:** Stability of SCFA after extraction and derivatization. Concentrations and variation percentages of 6 short chain fatty acids (acetate, propionate, isobutyrate, butyrate, isovalerate, valerate) quantified on the fresh and lyophilized stool of a unique human stool before and after 7 days of storage of the extract at +4 °C measured in triplicate (*n* = 3). AA: acetate; PA: propionate; BA: butyrate; IBA: isobutyrate; IVA: isovalerate; VA: valerate.

	AA	PA	BA	IBA	IVA	VA
Mean concentrations in lyophilized stool before storage	73.29	30.55	42.92	7.84	8.29	20.27
Mean concentrations in lyophilized stool after storage	69.14	26.66	39.57	7.01	7.72	20.88
Mean variation percentages in lyophilized stool after storage	−6%	−13%	−8%	−11%	−7%	3%
Mean concentrations in fresh stool before storage	25.55	8.56	9.63	1.66	2.02	4.28
Mean concentrations in fresh stool after storage	23.77	7.39	8.86	1.62	2.05	4.40
Mean variation percentages in fresh stool after storage	−7%	−14%	−8%	−2%	2%	3%

**Table 10 microorganisms-12-00828-t010:** Mean values, standard deviations, and ranges of each short chain fatty acid’s concentration in the control group versus the IBD group. CTRL: control group; IBD: intestinal bowel disease group; SD: standard Deviation.

SCFA (µmol/g of Stools)	CTRL	SD of CTRL	Range of CTRL	IBD	SD of IBD	Range of IBD
Acetate	151.2	84.3	22.6–301.6	163.7	94.0	45.1–364
Propionate	77.5	61.6	4.6–118.6	58.5	37.3	8.8–219
Isobutyrate	6.3	3.4	0.9–16	6.5	4.2	1–13.6
Butyrate	84.2	68.0	1.9–203.7	81.3	59.4	0.8–263.8
Isovalérate	8.2	4.8	1.3–11.5	6.5	3.5	1–20.3
Valérate	14.7	11.3	0.9–24.5	12.7	8.6	1.2–40.2

## Data Availability

Data are contained within the article.

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
