# Peer review of "Effect of Stool Sampling on a Routine Clinical Method for the Quantification of Six Short Chain Fatty Acids in Stool Using Gas Chromatography–Mass Spectrometry"

_microorganisms, 2024, doi:10.3390/microorganisms12040828_

Round 1

Reviewer 1 Report

Comments and Suggestions for Authors

Dear Editor,

Thank you so much for inviting me to review the article written by Dr Mahdi Tarek entitled “Impact of Stool Sampling on a Clinical Routine Method for the Quantification of Six Short-Chain Fatty Acids in Stool by Gas Chromatography-Mass Spectrometry”.

Authors mentioned that the important of the Short-chain fatty acids for our healthy, and the link between the SCFA and some human pathological conditions, such as inflammatory bowel disease, irritable bowel syndrome, diarrhea, and cancer; so it is essential and important to find a flexible method to detect them, and now the authors shows optimized detection for SCFA clinical routine method, make contribution to the diversity clinical laboratory purpose, and also make a more robust analytical method.

Finally the method the pre- and post-analytical comparison studies that could be easily used for clinical laboratory purposes and flexible to fit this purpose.

In my opinion, the method performed a large body of details for this experiment to demonstrate their conclusion. Their language is good, and structure of the manuscript is very organized. I have few comments as below:

(1)  In the line 144, the” 2.3. SCFA Extraction and Derivatization Method” section, the “1mL”should be “1ml” .

(2) In the line 197, 2.5.3 “Precision and Accuracy” section ,the “StabilityTo evaluate” should be make some modifications.

(3) In result” 3.3. Impact of Lyophilization and Glycerol Addition” section, why get the equivalent or higher concentrations of all measured SCFAs in lyophilized stools compared with fresh stools, and what’s the meaningful for this result. 

(4) In the line 302, result 3.3 section, in figure 3A , and the line 377, result 3.6 section, the“+4°c.” should be “+4.

(5) In the line 316, result 3.4 section, the “It is worth noting that “ should be make some modifications.

(6) In the line 333, result 3.4 section, table7 “(n=3).. “ should be”(n=3).”

(7) Since authors mentioned that this successfully validated method for the simultaneous determination of this six SCFAs, if there are any limitations for this method.

(8) If there are any other future improvements or optimization to this system.

Comments on the Quality of English Language

The language of the manuscript is ok, but moderate improvement should be made. 

Author Response

Thank you so much for inviting me to review the article written by Dr Mahdi Tarek entitled “Impact of Stool Sampling on a Clinical Routine Method for the Quantification of Six Short-Chain Fatty Acids in Stool by Gas Chromatography-Mass Spectrometry”.

Authors mentioned that the important of the Short-chain fatty acids for our healthy, and the link between the SCFA and some human pathological conditions, such as inflammatory bowel disease, irritable bowel syndrome, diarrhea, and cancer; so it is essential and important to find a flexible method to detect them, and now the authors shows optimized detection for SCFA clinical routine method, make contribution to the diversity clinical laboratory purpose, and also make a more robust analytical method.

Finally the method the pre- and post-analytical comparison studies that could be easily used for clinical laboratory purposes and flexible to fit this purpose.

In my opinion, the method performed a large body of details for this experiment to demonstrate their conclusion. Their language is good, and structure of the manuscript is very organized. I have few comments as below:

 We thank the reviewer for his kind comments.

(1)  In the line 144, the” 2.3. SCFA Extraction and Derivatization Method” section, the “1mL”should be “1ml”

This has been modified, thank you

(2) In the line 197, 2.5.3 “Precision and Accuracy” section ,the “StabilityTo evaluate” should be make some modifications.

This has been modified, thank you

(3) In result” 3.3. Impact of Lyophilization and Glycerol Addition” section, why get the equivalent or higher concentrations of all measured SCFAs in lyophilized stools compared with fresh stools, and what’s the meaningful for this result. 

This is discussed in the manuscript lines 321-324 (New file). It has been published before that in fresh stool (see for example, the work of Ueyama et al.) all SCFAs decrease rapidly with time if not frozen at -80°C. Even at 4°C, they undergo a rapid volatilization and are metabolized by the gut microbiota which could explain the better preservation with lyophilization.

 (4) In the line 302, result 3.3 section, in figure 3A , and the line 377, result 3.6 section, the“+4°c.” should be “+4℃”.

This has been modified throughout the manuscript, thank you

(5) In the line 316, result 3.4 section, the “It is worth noting that “ should be make some modifications.

This has been modified, thank you

 (6) In the line 333, result 3.4 section, table7 “(n=3).. “ should be”(n=3).”

This has been modified, thank you

 (7) Since authors mentioned that this successfully validated method for the simultaneous determination of this six SCFAs, if there are any limitations for this method.

We did not discuss the limitations of the method as we based this method on a previous work that we only slightly modified. Hence the limitations of our methods are essentially the same as the work from Furuhashi et al. The originality of the work is based on the comparison between the different options in stool management.

(8) If there are any other future improvements or optimization to this system.

There probably won’t be major future improvement in the analytical part of this project from our side, as this was never the aim of this project. But surely with time, others will find ways to improve the method. 

Reviewer 2 Report

Comments and Suggestions for Authors

A good, methodical work that can provide practical information for researchers in the field of understanding microbiota hemostasis. The moderate scope results from the narrow topic, but microbiota research has been gaining significantly recently.

Author Response

A good, methodical work that can provide practical information for researchers in the field of understanding microbiota hemostasis. The moderate scope results from the narrow topic, but microbiota research has been gaining significantly recently.

We thank the reviewer for his kind comments.

Reviewer 3 Report

Comments and Suggestions for Authors

The study performed by Tarek et al. focused on the development and validation of a targeted method for short-chain fatty acids (SCFAs) in human fecal samples. These metabolites are highly important gut microbiota-derived products that present significant roles in human health and disease. Therefore, the development of methods for accurately quantifying these metabolites in biological samples, particularly in feces, is desirable, and the study of the different storage conditions that could affect the integrity of the sample too. However, the paper is chaotic and far from being well written, and besides there are several points to be resolved/clarified:

As the main comment, if the objective is to standardize methods for future applications as diagnostic tools, this objective was not achieved.  The method can not be considered adequately validated for SCFA quantification in stool samples including the three main SCFAs (Acetate, Propionate, Butyrate) as well as 3 minor ones (Isobutyrate, Valerate, Isovalerate). Moreover, the novelty of their result is unsound, as they could not improve the current  “state of the art” for the application in clinics with other kinds of stool collection methods, such as RNA later of Methanol/ethanol solutions.

1.       ABSTRACT: the first part is general and retrieved from literature and not from their study. The derivatization method and general description of the study should be included

2.       In section 1 (Introduction), please include at least one reference following the sentence from lines 73 to 75.

3.       In section 1 (Introduction), consider revising the part from lines 84 to 93. Derivatization is required when complex samples are measured by GC-MS and GC-FID, but according to the text, it seems that is only for GC-MS. Besides, the current text gives the impression to the reader that LC-MS is the most suitable analytical technique for SCFAs analysis, thus prompting the question about the reasons behind selecting GC-MS for the development of the method.

4.       In section 1, the sentence covering lines 97 to 98 must be rewritten for clarification and better comprehension. It is hard to follow.

5.       In section 2.  The three internal standards (IS) were propionic acid d5, butyric acid d8, and 3-methylpentanoic acid but which one was for each analyte and why?. Besides, why stable isotopically labeled acetic acid was not included as IS for acetic acid?

6.       In section 2.2, All fecal samples were frozen at -80°C for a minimum of 24 h, then subjected to freeze-drying. So, how did they compare the different storage conditions?

7.       In section 2.3, the units for the concentration should be expressed as μmol/g of lyophilized/dried feces. Besides, how did they optimize the method?

8.       In section 2.4, the injection mode (split or splitless) should be indicated.

9.       In section 2.5.1, the specific steps where the QC feces were spiked to calculate the extraction recovery % should be indicated. Additionally, the explanation of the calculation for the matrix effect is not clear, the step where the QC feces were spiked should be indicated. Moreover, equation 2 is expressed as “extraction recovery %”, as equation 1, instead of “matrix effect %”. Equation 2 itself is unclear, as the matrix effect is calculated as follows: (concentration of the matrix spiked – endogenous concentration of the matrix) / concentration in solvent. This point should be carefully revised.

10.   In section 2.5.2, the IS used to calculate the response factor should be indicated, was the same IS for all the SCFAs? Did the authors use d5-propionic acid for C2-C3 SCFAs and d8-butyric acid for C4-C5 SCFAs?

11.   In section 2.5.3, the storage conditions and stability tests (from lines 199 to 221) should be presented in a separate section apart from “Precision and Accuracy”. Moreover, when comparing the storage conditions, how do the authors normalize or correct for the dilution of the sample when comparing fresh and diluted feces? Why only one stool sample was selected for the whole study? A pool sample per group would represent with more reliability the average composition of each.

12.   In section 2.5.3, why did the authors discard the storing condition at room temperature for lyophilized feces?

13.   In section 2.5.3, the stability tests proposed allow to compare a wide variety of storage conditions. Nevertheless, including a table or diagram would allow the reader to see more clearly the conditions being compared.

14.   In section 3 (Results and Discussion), the metabolite extraction and derivatization protocol should be discussed. This protocol involves numerous steps, increasing the risk of introducing bias. The metabolite concentration step (addition of chloroform and subsequent extraction of the aqueous upper phase for derivatization) could be avoided since the concentration of SCFAs in feces is not low. Additionally, metabolite extraction using isobutanol 10% in water at the beginning of the protocol instead of NaOH 0.2 M can lead to potential losses. Given that the initial steps aim to extract the metabolites from organic matter, introducing NaOH 0.2 M (pH around 12) at this point would enhance the solubility of SCFAs. This adjustment would ensure that SCFAs are in their unprotonated form (since the pKa of the targeted analytes ranges between 3 and 5), thereby facilitating the extraction of a higher quantity. The justification and relevance of these aspects should be included in the text, as well as the discussion comparing to other published methods.

15.   In section 3.1. This part should be rewritten, it is difficult to follow. Table 3. Linearity 1/x, but with IS calibration, what does it mean? Moreover, according to the introduction part: 60:20:20 (acetate:propionate:butyrate), so why the linearity range does not match with this ratio? Why the LOQ for butyric acid is so high compared to propionic acid? Was the LLOQ studied in terms of precision and accuracy? Why the spiked concentrations were those? Validation was not adequately designed. The concept of matrix effect should be revised. In GC-MS, the matrix effect determines the interaction of the analytes with active sites (e.g. the liner, transfer line, GC column, …), therefore it does not evaluate the effect of ionization.

16.   In section 3.1, the hypothesis explaining the differences in the extraction process efficiency observed between acetate, propionate and butyric, and the remaining SCFAs should be further explained, the risk of overestimation is due to higher endogenous concentrations? Was this issue previously reported?

17.   The results obtained for the method validation (section 3.1) should be compared to other methods previously published, highlighting the advantages and disadvantages of the method proposed by the authors. Literature recommendations for discussion: 10.1016/J.CHROMA.2021.462680, 10.1016/J.JCHROMB.2023.123930, 10.1016/J.AB.2017.12.001.

18.   Tables 2 and 3 could be merged, specifying the endogenous concentration, the spiked concentration, and which values correspond to inter-day and intra-day precision.

19.   In Table 5, “ionization matrix effect” should be replaced with “matrix effect” (explanation indicated in point 15). Furthermore, the equation used for the extraction process efficiency calculation is not indicated in the materials and methods, please include this information.

20.   In section 3.2, did the authors find any possible explanation for the increased SCFAs concentration in feces stored with methanol compared to fresh or other solvents?

21.   In section 3.3, did the authors take into account the percentage of dried mass between fresh and lyophilized feces to analyze the same relative amount? If not and the same amount of fresh and lyophilized feces was derivatized, this could explain the increased concentration of analytes in lyophilized feces. Otherwise, did the authors find any explanation for this phenomenon?

22.   In the results concerning the storage conditions and stability tests (Sections 3.3-3.6), is there a threshold for variation percentage? The authors should discuss why these values are acceptable.

23.   Table 6: what only the mean is reported? What about the SD/SEM?

24.   The numbering of the tables and figures should be revised, as they should be numbered in the order in which they appear in the text (for example table 9 before table 7)

25.   In the whole manuscript, please indicate the abbreviations in the table/figure captions.

26.   The manuscript text should be revised to correct redundancies and grammatical errors.

27.   Ref. 19, incomplete information of a book reference.

Comments on the Quality of English Language

English should be polished 

Author Response

The study performed by Tarek et al. focused on the development and validation of a targeted method for short-chain fatty acids (SCFAs) in human fecal samples. These metabolites are highly important gut microbiota-derived products that present significant roles in human health and disease. Therefore, the development of methods for accurately quantifying these metabolites in biological samples, particularly in feces, is desirable, and the study of the different storage conditions that could affect the integrity of the sample too. However, the paper is chaotic and far from being well written, and besides there are several points to be resolved/clarified:

We thank the reviewer for his comments. As the views on the paper’s organization and writing are diametrically opposite between reviewer 1 and 3, we will leave this matter in the hands of the editor.

As the main comment, if the objective is to standardize methods for future applications as diagnostic tools, this objective was not achieved.  The method can not be considered adequately validated for SCFA quantification in stool samples including the three main SCFAs (Acetate, Propionate, Butyrate) as well as 3 minor ones (Isobutyrate, Valerate, Isovalerate). Moreover, the novelty of their result is unsound, as they could not improve the current  “state of the art” for the application in clinics with other kinds of stool collection methods, such as RNA later of Methanol/ethanol solutions.

We would like to emphasize that this study was not aimed at improving analytical performances for SCFA detection, nor was it to find a way to optimize SCFA quantitation in different collection methods to standardize them. What is novel is that we compared, with one robust analytical methods (a slightly modify method based on Furuhashi et al.), the results from different “state of the art” conditions for stool management, because, to our knowledge, it had never been done before. The idea is then to use the results from this study to standardize good practices (i.e preferentially use lyophilization when possible) between clinical laboratory so that clinicians and scientist alike can get the best of their pre-analytical conditions and at least know the biases when the best option is unavailable.

  1. ABSTRACT: the first part is general and retrieved from literature and not from their study. The derivatization method and general description of the study should be included

We agree that the first part is rightfully general, aimed at introducing the reader to the subject. We added the derivatization method and general description as well as some precision regarding our conclusion.

  1. In section 1 (Introduction), please include at least one reference following the sentence from lines 73 to 75.

This has been modified, Thank you

  1. In section 1 (Introduction), consider revising the part from lines 84 to 93. Derivatization is required when complex samples are measured by GC-MS and GC-FID, but according to the text, it seems that is only for GC-MS. Besides, the current text gives the impression to the reader that LC-MS is the most suitable analytical technique for SCFAs analysis, thus prompting the question about the reasons behind selecting GC-MS for the development of the method.

This has been modified, Thank you

  1. In section 1, the sentence covering lines 97 to 98 must be rewritten for clarification and better comprehension. It is hard to follow.

This has been modified, Thank you

  1. In section 2.  The three internal standards (IS) were propionic acid d5, butyric acid d8, and 3-methylpentanoic acid but which one was for each analyte and why?. Besides, why stable isotopically labeled acetic acid was not included as IS for acetic acid?

       Thank you for this comment, we added this information in the paper in a new Table 1. Propionic acid D5 was used for propionic acid, Butyric acid D8 for butyric acid and isobutyric acid. We indeed used deuterated acetic acid for acetic acid, this information was overlooked when writing the manuscript. 3-methypropanoic acid was used for the remaining SCFAs. Please note that in his study, Furuhashi uses  only one IS (3-methypropanoic acid) and still obtains comparable results. Nevertheless, we thought that it would be safer to use as many as available.

  1. In section 2.2, All fecal samples were frozen at -80°C for a minimum of 24 h, then subjected to freeze-drying. So, how did they compare the different storage conditions?

       Thank you for this comment. It is right that this section of the methods doesn’t make sense regarding the results of the study. Wen this part was written, it was in reference to the “normal” preanalytical condition, which, to us, is lyophilization. We modified this section to better represent what has been done in the study.

  1. In section 2.3, the units for the concentration should be expressed as μmol/g of lyophilized/dried feces. Besides, how did they optimize the method?

Thank you, this has been modified.

  1. In section 2.4, the injection mode (split or splitless) should be indicated.

Thank you, this has been added to the section 2.4 : Split  50:1

  1. In section 2.5.1, the specific steps where the QC feces were spiked to calculate the extraction recovery % should be indicated. Additionally, the explanation of the calculation for the matrix effect is not clear, the step where the QC feces were spiked should be indicated. Moreover, equation 2 is expressed as “extraction recovery %”, as equation 1, instead of “matrix effect %”. Equation 2 itself is unclear, as the matrix effect is calculated as follows: (concentration of the matrix spiked – endogenous concentration of the matrix) / concentration in solvent. This point should be carefully revised.

       Thank you, we indicated the precise steps where QC feces where spiked (section 2.3) and there was indeed a copy-paste error in equation N°2 which has been modified.

  1. In section 2.5.2, the IS used to calculate the response factor should be indicated, was the same IS for all the SCFAs? Did the authors use d5-propionic acid for C2-C3 SCFAs and d8-butyric acid for C4-C5 SCFAs?

       This was done as for the method, which we detailed in point 5 of the reviewer’s comments.

  1. In section 2.5.3, the storage conditions and stability tests (from lines 199 to 221) should be presented in a separate section apart from “Precision and Accuracy”. Moreover, when comparing the storage conditions, how do the authors normalize or correct for the dilution of the sample when comparing fresh and diluted feces? Why only one stool sample was selected for the whole study? A pool sample per group would represent with more reliability the average composition of each.

       Thank you. A stability section was added. Only one feces was used as there is great interindividual variability among stool samples and we thought it would limit the manipulation of the sample and thus the degradation/volatility of the SCFAs.

  1. In section 2.5.3, why did the authors discard the storing condition at room temperature for lyophilized feces?

       The reviewer is right, we could have done it as well, but Ueyama et al. showed that there is no difference in stability between lyophilized feces at room temperature and crude feces at -80°C.

  1. In section 2.5.3, the stability tests proposed allow to compare a wide variety of storage conditions. Nevertheless, including a table or diagram would allow the reader to see more clearly the conditions being compared.

       We considered doing as per reviewer recommendation, but we felt that the paper had enough tables already (hence why we fused 4 of them into two) and that this would make it heavier.

  1. In section 3 (Results and Discussion), the metabolite extraction and derivatization protocol should be discussed. This protocol involves numerous steps, increasing the risk of introducing bias. The metabolite concentration step (addition of chloroform and subsequent extraction of the aqueous upper phase for derivatization) could be avoided since the concentration of SCFAs in feces is not low. Additionally, metabolite extraction using isobutanol 10% in water at the beginning of the protocol instead of NaOH 0.2 M can lead to potential losses. Given that the initial steps aim to extract the metabolites from organic matter, introducing NaOH 0.2 M (pH around 12) at this point would enhance the solubility of SCFAs. This adjustment would ensure that SCFAs are in their unprotonated form (since the pKa of the targeted analytes ranges between 3 and 5), thereby facilitating the extraction of a higher quantity. The justification and relevance of these aspects should be included in the text, as well as the discussion comparing to other published methods.

       All the necessary tests were already conducted by Furuhashi et al. and discussed in their paper. We don’t feel there is a need to add more as their study is well cited and referenced in our manuscript. Once again, we didn’t aim at publishing a more sensitive method than what was already published, but simply we wanted to compare the stool management processes in the preanalytical phase. We changed minor points compared to Furuhashi’s work (sample mass, IS, vortexing, hexane is substituted with cyclohexane and we used a slightly different GC-MS method) but still decided to validate the method again, partly because most of his work was not done on human feces.

  1. In section 3.1. This part should be rewritten, it is difficult to follow.

 Table 3. Linearity 1/x, but with IS calibration, what does it mean?

We are sorry, as much as we would like to comply, we fail to understand the question the reviewer is asking.

Moreover, according to the introduction part: 60:20:20 (acetate:propionate:butyrate), so why the linearity range does not match with this ratio?

The original article from which we collected this information states “Acetate, propionate, and butyrate are present in an approximate molar ratio of 60:20:20 in the colon and stools”. It is approximate, and we believe that the ratio in our data is close enough from this ratio as the mean of the three stools in table 3 is 68 µmol/g,19 µmol/g, 26 µmol/g for acetate, propionate and butyrate respectively, which gives us a ratio of 60:17:23.

Why the LOQ for butyric acid is so high compared to propionic acid?

       Because the level of response from the MS is better for propionic acid as shown by LOD. But this is without consequences on the study as levels of these analytes in samples are usually high enough and we are not limited by the quantity of stool sample available.

Was the LLOQ studied in terms of precision and accuracy?

       Again, we felt that the core of the method being already published, this was beyond the scope of our study.

Why the spiked concentrations were those?

The lower concentration used corresponds to the highest of the LOQs (butyrate) of the method. The higher concentration corresponds to an average of values for SCFAs found in Reygner et al, one of our previous study.

Validation was not adequately designed. The concept of matrix effect should be revised. In GC-MS, the matrix effect determines the interaction of the analytes with active sites (e.g. the liner, transfer line, GC column, …), therefore it does not evaluate the effect of ionization.

  1. In section 3.1, the hypothesis explaining the differences in the extraction process efficiency observed between acetate, propionate and butyric, and the remaining SCFAs should be further explained, the risk of overestimation is due to higher endogenous concentrations? Was this issue previously reported?

To our knowledge, this was not previously reported. Furuhashi shows a difference in recovery rates for butyrate as being app. 40% lower than acetate and propionate, bot doesn’t mention the less abundant SCFA.

  1. The results obtained for the method validation (section 3.1) should be compared to other methods previously published, highlighting the advantages and disadvantages of the method proposed by the authors. Literature recommendations for discussion: 10.1016/J.CHROMA.2021.462680, 10.1016/J.JCHROMB.2023.123930, 10.1016/J.AB.2017.12.001. 

Again, we think that this comparison is out of the scope of this work as the core method was already published elsewhere.

  1. Tables 2 and 3 could be merged, specifying the endogenous concentration, the spiked concentration, and which values correspond to inter-day and intra-day precision.

Thank you, this has been modified. We merged table 1 with table 2 and table 3 with table 4

  1. In Table 5, “ionization matrix effect” should be replaced with “matrix effect” (explanation indicated in point 15). Furthermore, the equation used for the extraction process efficiency calculation is not indicated in the materials and methods, please include this information.

Thank you, this has been modified. After reading the reviewer’s comment, we decided that “extraction process efficiency” is somewhat redundant with recovery and decided to exclude it from the manuscript.

  1. In section 3.2, did the authors find any possible explanation for the increased SCFAs concentration in feces stored with methanol compared to fresh or other solvents?

It has been published previously that in fresh stool (see for example, the work of Ueyama et al.) all SCFAs decrease rapidly with time if not frozen at -80°C. Even at 4°C, they undergo a rapid volatilization and are metabolized by the gut microbiota, so it is possible that degradation/stability or SCFA is different depending on the solvent, but this remains a hypothesis.

  1. In section 3.3, did the authors take into account the percentage of dried mass between fresh and lyophilized feces to analyze the same relative amount? If not and the same amount of fresh and lyophilized feces was derivatized, this could explain the increased concentration of analytes in lyophilized feces. Otherwise, did the authors find any explanation for this phenomenon?

       Yes, this was taken into account. In a previous publication (Reygner et al), we showed that SCFA were also better conserved when lyophilized compared to classic -20°C conservation methods. This could be explained by higher volatility/degradation.

  1. In the results concerning the storage conditions and stability tests (Sections 3.3-3.6), is there a threshold for variation percentage? The authors should discuss why these values are acceptable.

       We used +/- 20% as a threshold. This has been added to the manuscript in section 3.1

  1. Table 6: what only the mean is reported? What about the SD/SEM?

Thank you, this has been added to table 6 (new table 4).

  1. The numbering of the tables and figures should be revised, as they should be numbered in the order in which they appear in the text (for example table 9 before table 7)

Thank you, this has been done throughout the manuscript

  1. In the whole manuscript, please indicate the abbreviations in the table/figure captions.

Thank you, this has been checked and modified throughout the manuscript

  1. The manuscript text should be revised to correct redundancies and grammatical errors.

Although we had done it already, the manuscript has been sent out again for English language checking by “Cambridge proofreading”. We keep the certificate at the editor’s disposal.

  1. Ref. 19, incomplete information of a book reference.

Thank you, this has been modified.

Round 2

Reviewer 1 Report

Comments and Suggestions for Authors

Dear authors,

THank you so much for your revisions about my questions, and I endorsed the manuscript. 

Br,

Comments on the Quality of English Language

Minor revision

Author Response

We thank Reviewer1 for his kind comments and endorsement of our manuscript